# Dual-Band/Dual-Mode Rat-Race/Branch-Line Coupler Using Split Ring Resonators

**Mohammad Saeid Ghaffarian [1], Gholamreza Moradi [2], Somayyeh Khajehpour [1], Mohammad Mahdi Honari [1] and Rashid Mirzavand [1,*]**

[1] Intelligent Wireless Technology Laboratory, Electrical and Mechanical Engineering Department, University of Alberta, 9211 116 Street NW, Edmonton, AB T6G 1H9, Canada; msghaffarian@ualberta.ca (M.S.G.); skhajehp@ualberta.ca (S.K.); honarika@ualberta.ca (M.M.H.)

[2] Wave Propagation and Microwave Measurement Research Laboratory, Department of Electrical Engineering, Amirkabir University of Technology, Tehran 15914, Iran; ghmoradi@aut.ac.ir

* Correspondence: rmirzavand@ualberta.ca

**Abstract:** A novel dual-band/dual-mode compact hybrid coupler which acts as a dual-band branch-line coupler at the lower band and as a rat-race coupler at the higher band is presented in this paper. One of the most interesting features of the proposed structure is that outputs of the proposed coupler in each mode of operation are on the same side. This unique design is implemented using artificial transmission lines (ATLs) based on open split ring resonators (OSRR). The low-cost miniaturized coupler could be operated as a dual-band 90° branch-line coupler at 3.3 and 3.85 GHz and 180° rat-race coupler at 5.3 GHz. The proposed coupler could be utilized in the antenna array feeding circuit to form the antenna beam. The structure's analytical circuit design based on its equivalent circuit model is provided and verified by measurement results.

**Keywords:** artificial transmission line; branch-line coupler; dual-band; dual-mode; feeding circuit; open split ring resonator; phased array antenna; rat-race coupler

## 1. Introduction

Artificial transmission lines with right/left hand wave propagation characteristics and multi-band functionality can be implemented by loading a host line with shunt inductances and series capacitances [1–3], split ring resonators (SRR) or complementary split ring resonators (CSRRs) [4–7]. The artificial transmission lines have fantastic characteristics such as the controllability of the dispersion diagram, characteristic impedance and phase of the transmission line, compared to the conventional transmission lines, the dual-band or multi-band structures could be realized. The dual-band coupler consists of transmission lines with identical characteristic impedance and specific phase shifts at the desired frequencies. In [1], dual-band quarter-wave composite right–left-handed transmission lines (CRLH TL) with −90° and −270° phase shifts were presented at the operation frequency band. The proposed CRLH TLs in [1] have the same phase response as conventional couplers and are larger than typical microstrip couplers. By considering the phase delay and the use of CRLH TLs in dual bands, the dual-band directional couplers are implemented in [2]. A dual-mode coupler can operate identically to either a branch-line coupler (90°) or a rat-race (180°) at specific frequencies, as demonstrated in [3]. The size of the structure was identical to a conventional branch-line coupler and could operate at only two different frequency bands. A brief design of dual-mode/dual-band rat-race and branch-line coupler was also presented in [4]. The implementation of dual-band and multi-band couplers by means of artificial TLs based on CSRRs and SRRs was introduced in [5–7]. The realization of a dual-band coupler with an octave bandwidth based on the resonant type of metamaterial (CSRR) was shown in [5]. The quad-band coupler using SRR, meander inductors, patch and interdigital capacitors was proposed in [6]. Its frequency bandwidth was very narrow,

and its insertion loss was higher than that of the conventional types. However, there are many single mode dual-band rat-race and branch line couplers in the literature. We briefly review some of them here. A dual-band patch hybrid coupler with arbitrary power division ratios was presented in [7,8]. Some other dual-band couplers were proposed based on the loading of the conventional couplers with stubs [9], short/open-ended stubs [10] or additional integrated coupling sections with conventional couplers [11,12]. A dual-band rat-race coupler based on a folded substrate integrated waveguide was proposed in [13]. By loading CRLH transmission lines into the structure, a miniaturized coupler was designed. A compact dual-band branch line coupler by using the π-shaped dual transmission lines at two arbitrary frequencies was presented in [14]. Another method for designing dual-band operation was based on impedance transformation that can create a structure with operation as a coupler as well as a phase shifter simultaneously [15]. Dual-band couplers with an arbitrary coupling coefficient by using a coupled lines configuration were proposed in [16–19]. A dual-band coupler with wide separation between frequency bands was designed by applying three cascade coupled sections in the coupler branches [20]. Reconfigurable dual-band coupler designs were also presented [21,22]. Another technique in designing compact couplers by using coupled resonators was introduced in [23]. However, by increasing the demand for small multi-band devices, multi-mode as well as multi-band couplers have an important role to achieve this goal in the electronics industry. The compactness can be realized by introducing new circuits with multi-mode functionality.

In this paper, a novel compact dual-band/dual-mode planar single layer directional coupler is presented. The proposed design acts as a dual-band hybrid coupler at 3.3/3.85 GHz and a rat-race coupler at 5.3 GHz. The TL branches of the proposed coupler are implemented by using a combination of OSRR unit cells. The designed resonant type metamaterial TLs have the capability to control phase response and characteristic impedance at the desired frequency bands. The phase slopes of the coupler branches are engineered until the proposed coupler functions identically to both a dual-band 90° hybrid coupler and a 180° rat-race coupler at the desired frequency band of operation. The outputs of the proposed coupler are designed to be on the same side. The proposed structure could be used in feeding networks in the different microwave and antenna arrays. The theoretical analysis and synthesis of OSRR TL are first performed then the simulation, and experimental results of the proposed coupler are demonstrated to validate the proposed approach.

## 2. Circuit Model and Implementation of Artificial Transmission by Means of OSSR

Figure 1 shows the schematic and dimensions of the dual-band/dual-mode proposed coupler. The structure includes four artificial TLs based on OSRR with two unit cells of OSRR in vertical branches (OSRR1 and OSRR2) and one unit cell in horizontal (OSRR3) sections. Using the OSRR, the proposed coupler functions as a dual-band branch line coupler in the lower bands and reconfigures identically to a rat-race coupler in the high band with the outputs at the same side (Figure 1). The metamaterial branches in horizontal and vertical sections have specific characteristic impedance and phase responses, based on the specific mode of operation. The artificial transmission lines are implemented on 0.8 mm thick RO4003C substrate with $\varepsilon_r$ = 3.55 and tanδ = 0.002. The desired operating frequencies are $f_{L1}$ = 3.05 and $f_{L2}$ =3.6 GHz for the branch-line coupler and $f_H$ = 5.5 GHz for the rat-race coupler. These frequency bands are selected to cover wireless applications such as WLAN and LTE. The design process and circuit model of the artificial TLs and the functionality of each mode are illustrated in detail as follows.

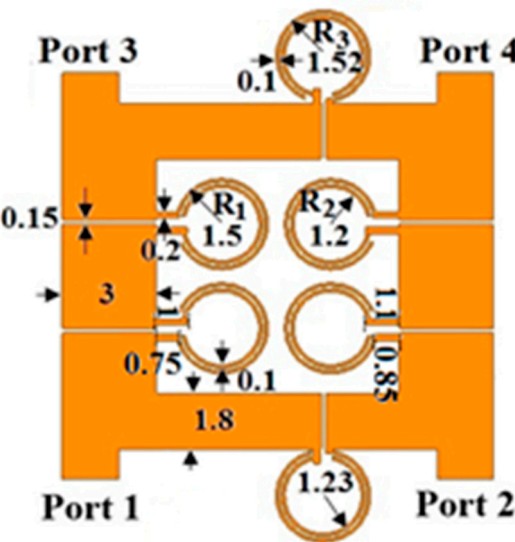

**Figure 1.** Topology of designed dual band/dual mode coupler (all dimensions are in mm).

### 2.1. Rat-Race Coupler

The conventional rat-race coupler with 0° and 180° phase shifts at its outputs (sum ($\sum$) and delta ($\Delta$)) consists of three 90° sections and another section with a 270° phase shift. The characteristic impedance of the entire coupler is 70 $\Omega$ to provide equal power division. In most popular configurations, the coupler outputs are on different sides of the structure. Typically, in applications requiring outputs on the same side (such as feeding networks in phased array antennas or other microwave circuits [24–27]), the 3 dB hybrid coupler with a combination of 90° differential phase shifters [28–31] is used. The employing of the phase shifters increases the size of the structure and unequal loss in the output ports. In [32], a coupler with a nonstandard phase difference was proposed, based on different characteristic impedances and electrical lengths of different TL sections. However, the designed coupler could not be applied for 0 and 180° outputs phase shift. Employing lumped reactances between TLs to provide a realizable characteristic impedance and shorten the transmission line length was proposed in [33]. By using this idea, a rat-race coupler with harmonic suppression and arbitrary power division ratios was designed.

For the first time, a novel structure is proposed in this paper that has the capability to work like a rat-race coupler with outputs at the same side. The proposed configuration is shown in Figure 2a. The design includes four sections with identical horizontal branches (based on Figure 1 that there is coupling between vertical branches) and the vertical sections with the same characteristic impedance and different electrical lengths. The novel idea that leads to producing a rat-race coupler with the functionality on the same side outputs (port 3 and port 4), is the implementation of a coupling inductor between the vertical sections of the proposed rat-race coupler. Usually, with a structure like a hybrid coupler, even by using different characteristic impedances or electrical lengths, the rat-race operation could not be obtained. However, by using the proposed structure, the rat-race coupler in different frequencies could be realized. Design equations and constraints can be derived by even and odd mode analysis [34]. The design constraints can be introduced, based on good reflection and isolation coefficients, the constant phase at the outputs and desired power division ratio. Since the unknown design parameters of the equivalent circuit model (Figure 2a) are seven and we have only four design equations (based on S-parameters in odd and even mode analysis), thus the possibility of determining circuit parameters in the extraction method would be complicated. The design parameters of the proposed circuit model were achieved based on manual curve fitting.

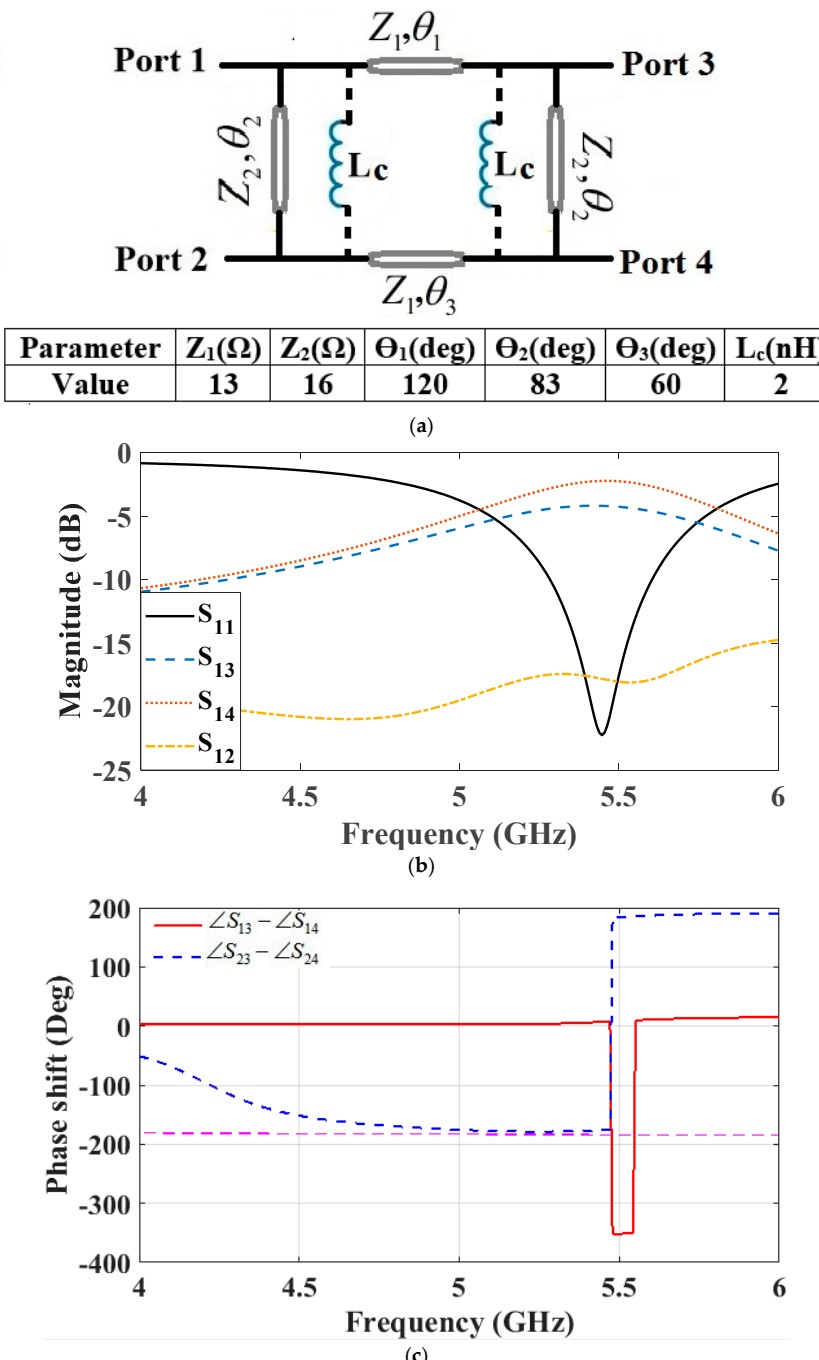

**Figure 2.** (**a**) The transmission line model of the proposed hybrid rat-race coupler, (**b**) the amplitude and (**c**) phase response of the proposed coupler.

However, we illustrated the results of the circuit model with the values of the circuit model parameters to verify the functionality of the proposed structure. Although the inserted inductance can be realized by different approaches, such as using lumped elements or utilizing coupling of two conductors in the microstrip technology, the method selection depends on the application and implementation capabilities of various sections of the proposed coupler. The obtained parameters to design a rat-race coupler at 5.5 GHz center frequency with the outputs at the same side are illustrated in Figure 2a. The S-parameter amplitude response of the designed coupler and the phase difference between outputs (port 3 and port 4) and sum port (port 1) and difference port (port 2) are shown in Figure 2b,c, respectively. As shown, the desired performance, such as a good return

loss, isolation, and proper phase difference (178° and 5° phase shift between the outputs), was achieved.

### 2.2. Branch Line Coupler

The conventional branch line coupler has two TLs in the vertical and horizontal branches with 35 Ω and 50 Ω characteristic impedances and +90° electrical length. Therefore, the equivalent artificial TLs of the proposed coupler at $f_{L1}$ = 3.05 and $f_{L2}$ = 3.6 GHz should have the same characteristics ($Z_1$ = 50 Ω, $Z_2$ = 35 Ω, $\theta_{L1}$ = ±90° and $\theta_{L2}$ = ±90°). Therefore, we had four conditions that were used to determine the proposed coupler equivalent circuit in the branch line mode coupler. The next step was to design the metamaterial TLs sections based on the design constraints discussed in the previous subsections.

### 2.3. Artificial TLs Design

The open split ring resonators (OSRRs) are introduced in [35] for the first time. As mentioned in [35], we could assume the equivalent circuit model of the proposed OSRR to be as depicted in Figure 3. The host transmission line characteristic impedance, electrical length, and inductance value, $L_m$, of the OSRR are the same as SRR with identical dimensions. However, the circuit model capacitance, $C_m$, of the OSRR is four times the capacitance of the corresponding SRR. Therefore, the resonance frequency of the structure based on OSRRs is half the one designed by using SRR resonators. As shown in [6], the mentioned series $LC$ equivalent circuit model could not characterize the properties of OSRRs properly. There are different ring resonator coupling methods which can be used in various applications [36]. The SRR resonators are connected directly to the microstrip TLs. The transmission line loaded by an OSRR in microstrip technology shows an additional phase shift due to having $Lp$ and $C_p$ in the equivalent circuit model. Figure 3 demonstrates the OSRR unit cell with its equivalent circuit model. The microstrip transmission lines connected to both sides of the OSRRs are served as the right-handed (RH) part, and the OSRR equivalent circuit model can be assumed as the left-handed (LH) part.

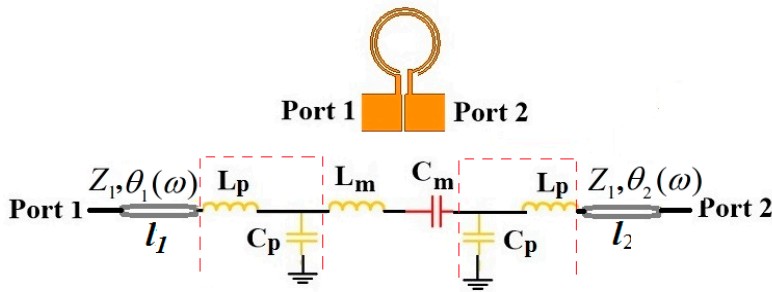

**Unit: L: nH, C: pF, Z: Ω, l: mm**

| Parameter | $L_P$ | $C_P$ | $L_m$ | $C_m$ | $Z_1$ | $l_1$ | $l_2$ | |
|---|---|---|---|---|---|---|---|---|
| Value | 1.9 | 0.4 | 10 | 0.2 | 35 | 2.7 | 2.7 | For Vertical TL with 2 OSRR |
| Parameter | $L_P$ | $C_P$ | $L_m$ | $C_m$ | $Z_1$ | $l_1$ | $l_2$ | |
| Value | 2.9 | 0.4 | 7.5 | 0.16 | 50 | 4 | 5.2 | For Horizontal TL with 1 OSRR |

**Figure 3.** Circuit model (unit cell) of proposed artificial TLs based on OSRR with obtained elements value for vertical and horizontal sections.

To simplify the structure analysis in this step, we do not consider parasitic elements for calculating the circuit model parameters. Therefore, in branch line mode, the vertical artificial TLs should be equivalent to a 35 Ω/−90° at $f_{L1}$ = 3.05 GHz and 35 Ω/ 90° at $f_{L2}$ = 3.6 GHz, and a 13 Ω in the rat- race coupler mode at 5.5 GHz. As the other vertical TL has a different electrical length, this TL is not considered here, for the sake of brevity. It should be noted that the coupling inductance in the rat-race coupler design has a direct effect on the differential phase shift between its outputs. Therefore, we consider the

characteristic impedance equation at the high-frequency band. However, the horizontal section can be 50 $\Omega/-90°$ at $f_{L1}$ = 3.05 GHz and 50 $\Omega/90°$ at $f_{L2}$ = 3.6 GHz and 16 $\Omega$ at $f_H$ = 5.5 GHz center frequency. The ABCD matrix of the proposed artificial TLs unit cell should be identical to that of the ABCD matrix of a transmission line with $Z_c(\omega), \theta_c(\omega)$ characteristics. By equating the corresponding of these two matrix elements, the characteristic impedance and the phase response of TLs with OSRRs unit cell as a function of frequency can be obtained as,

$$\cos(\theta_c(\omega)) = A(\omega) \tag{1}$$

$$Z_c = \sqrt{\frac{B(\omega)}{C(\omega)}} \tag{2}$$

and the ABCD matrix of the artificial TL, which is loaded by an OSRR, is defined as follows:

$$A(\omega) = \cos(\theta_1(\omega))\cos(\theta_2(\omega)) + (\tfrac{j}{Z_1}\sin(\theta_2(\omega))(\cos(\theta_1(\omega))(j\omega(L_m - C_m)$$
$$-jZ_1\sin(\theta_1(\omega)))) \tag{3}$$

$$B(\omega) = \cos(\theta_2(\omega))(\cos(\theta_1(\omega))(j\omega(L_m - C_m)) + jZ_1\sin(\theta_1(\omega)))$$
$$+jZ_1\cos(\theta_1(\omega))\sin(\theta_2(\omega)) \tag{4}$$

$$C(\omega) = \tfrac{1}{Z_1}(\sin(\theta_2(\omega))(\cos(\theta_1(\omega)) + (\tfrac{j}{Z_1}\sin(\theta_1(\omega))(j\omega(L_m - C_m)))))$$
$$+\tfrac{j}{Z_1}\sin(\theta_1(\omega))(\cos(\theta_2(\omega))) \tag{5}$$

We have five unknown design parameters ($Z_1$, $\theta_1$, $\theta_2$, $L_m$, $C_m$) and five constraints which are:

$$Z_c(f_{L1}) = Z_V(f_{L1}) = 35 \ \Omega \tag{6}$$

$$Z_c(f_{L2}) = Z_V(f_{L2}) = 35 \ \Omega \tag{7}$$

$$\theta_c(f_{L1}) = \theta_V(f_{L1}) = -90°/2 \tag{8}$$

$$\theta_c(f_{L2}) = \theta_V(f_{L2}) = 90°/2 \tag{9}$$

$$Z_c(f_H) = Z_V(f_H) = 13 \ \Omega \tag{10}$$

The vertical section parameters of the proposed coupler were numerically calculated as $L_m$ = 9 nH, $C_m$ = 0.15 pF, $Z_1$ = 54 $\Omega$, $\theta_1$ = 19° and $\theta_2$ = 19°. The initial parameters of the vertical section unit cell are obtained. The next step is determining other circuit parameters ($L_p$ and $C_P$). As mentioned previously, to simplify the ABCD matrix, these elements were neglected in the previous step.

Now, firstly, the initial layout of the RH section (microstrip TL at both sides) and LH part (OSRR) is achieved based on the obtained parameters by curve fitting between electromagnetic (EM) simulation (by using high-frequency simulation software (HFSS)) and circuit model simulation (by using advanced design system (ADS)). The first estimation of the OSRR unit cell could be achieved by converting the circuit model values to ring resonators, as mentioned in [37,38]. An optimization procedure with EM simulation is necessary to obtain the desired values from Equations (5)–(9). By considering the accurate circuit model of Figure 3 and performing some curve fitting between EM and circuit model simulations, the parasitic elements could be inferred. The EM simulation based on the desired response is achieved by using two-unit cells of Figure 3 in vertical sections while the horizontal sections are composed of the one-unit cell (Figure 1).

The values of the elements of the horizontal TLs equivalent circuit model are obtained with the same analysis but with different design equating as follow:

$$Z_c(f_{L1}) = Z_H(f_{L1}) = 50 \ \Omega \tag{11}$$

$$Z_c(f_{L2}) = Z_H(f_{L2}) = 50 \ \Omega \tag{12}$$

$$\theta_c(f_{L1}) = \theta_H(f_{L1}) = -90^\circ \tag{13}$$

$$\theta_c(f_{L2}) = \theta_H(f_{L2}) = 90^\circ \tag{14}$$

$$Z_c(f_H) = Z_H(f_H) = 16\ \Omega \tag{15}$$

By using mathematical tools, the horizontal section parameters of the proposed coupler were numerically calculated as $L_m$ = 7 nH, $C_m$ = 0.17 pF, $Z_1$ = 38 Ω, $\theta_1$ = 32°and $\theta_2$ = 52° with cells. A similar process as the vertical section was done for an accurate circuit model of other sections. It should be mentioned that the desired response in horizontal branches was obtained with a one-unit cell. In this step, we recalculated the equivalent circuit model of the vertical and horizontal section parasitic elements into the circuit model. The obtained optimized parameters are tabulated in Figure 3. The comparison between the circuit model simulation by ADS and the EM simulation results by HFSS is shown in Figure 4. Figure 4a shows the comparison of the amplitudes (reflection and transmission coefficients) and the phase responses of the artificial TLs, respectively. Reasonable agreement between the results was observed even though the insertion loss of the circuit model due to the lack of the loss was better than the EM simulation results. The corresponding phase response of the two simulation methods matched; hence, the design approach for metamaterial TLs was verified.

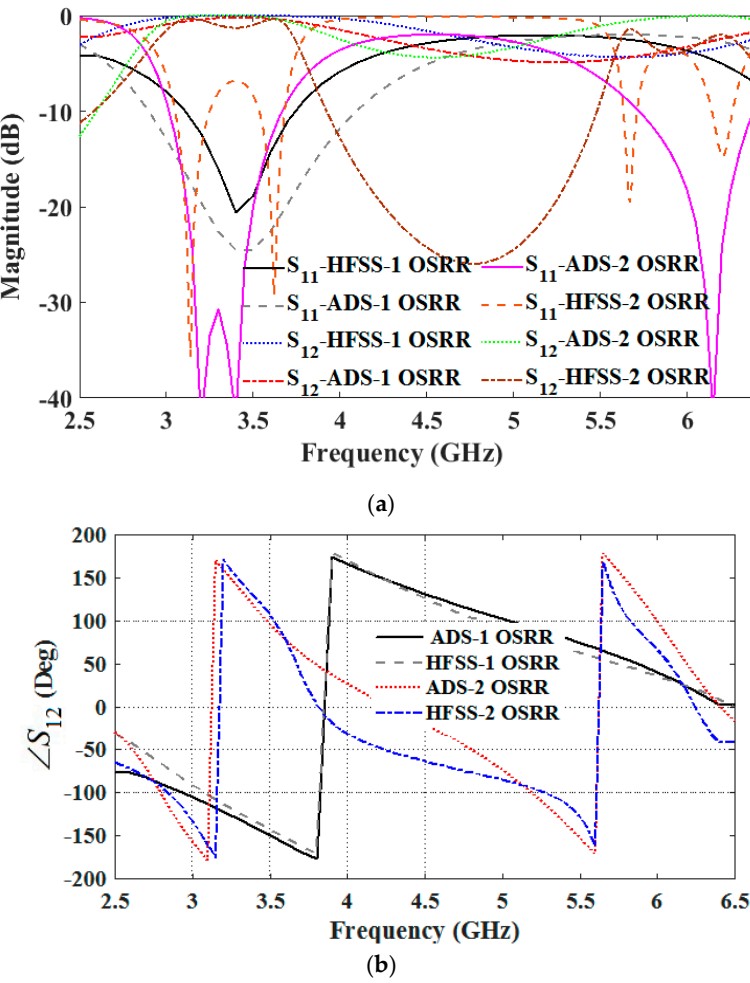

**Figure 4.** The amplitude and phase response of the proposed transmission lines circuit and EM simulations: (**a**) the magnitude of reflection and transmission coefficients and (**b**) the phase response of $S_{12}$.

*2.4. Accurate Circuit Model and Electromagnetic Simulation of the Proposed Coupler*

In the previous subsection, the equivalent circuit model of the desired dual-band/dual-mode coupler was obtained. The last step was to create a coupler with the combination of proposed artificial TLs. As shown in Figure 1, the two vertical sections with the OSRR were placed closely together, and due to mutual coupling effects, the desired results were achieved. The EM simulation results verified these effects in several ways. The coupling had a huge impact on the high frequency band in rat-race mode. As discussed before, with the inductance between the two parallel branches, the rat-race operation could be obtained. The magnetic currents between the resonators on the opposite side are coupled together, altering the amplitude and phase of the proposed coupler. This mutual coupling is modelled as a series parasitic capacitance ($C_c$) and simplified mutual inductance transformer introduced as ($L_C$) in the equivalent circuit model. Thus, the parameters' values of different resonators in the circuit model configuration should be optimized to achieve the best results. The elements of the circuit model were recalculated by curve fitting the electromagnetic simulation results and circuit model simulations. The obtained results showed that the proposed coupler operated as a dual-band branch line coupler at the center frequency of 3.05 and 3.6 GHz. Furthermore, the simulation results indicated that the proposed coupler acted like a rat-race coupler at the center frequency of 5.5 GHz, where sum and delta excitation ports are on the same side. A reasonable matching and isolation with 3 dB and 5 dB coupling factors in branch line and rat-race modes were achieved, respectively. The simulation results also indicated that the proposed coupler acted like a rat-race coupler at the center frequency of 5.5 GHz, where sum and delta excitation ports are on the same side. A reasonable matching and isolation with 3 dB and 5 dB coupling factors in branch line and rat-race modes were achieved, respectively. The further analysis is explained in the next section.

Figure 5a depicts the completed circuit model of the proposed coupler with the obtained values of the parameters. Based on Figure 2a, the proposed coupler vertical sections should have different electrical lengths to act as a rat-race coupler. This difference between branches is introduced in $L_{p1}$ and $C_{P1}$. The comparisons between the obtained results of the EM simulation and circuit model analysis are illustrated in Figure 5b,c. The phase differences between the outputs in the two simulations had the same trend, especially at the desired operating frequency despite the complexity of the circuit model and structure.

*2.5. Parametric Study of the Proposed Coupler*

To demonstrate the effects of the dimensions of the OSRRs on the performance of the proposed coupler, a parametric study (by HFSS) was done by changing the radius of the OSRR in horizontal and vertical sections. As shown in Figure 6a, by altering the length of the resonators, the resonance frequency was shifted due to the variation of the equivalent inductance. By exploiting this, the desired operating frequency could be achieved. Figure 6b,c illustrate the variation of the transmission coefficients versus the different radii of the ring resonators. As explained, the dimensions of the ring resonators play a key role in the best results at the operating frequency band of interest. The variation of the differential phase shifts between the outputs and sum port is depicted in Figure 6d. By changing the specification of the proposed coupler resonator, the obtained differential phase shift, as well as the operating frequency, were altered.

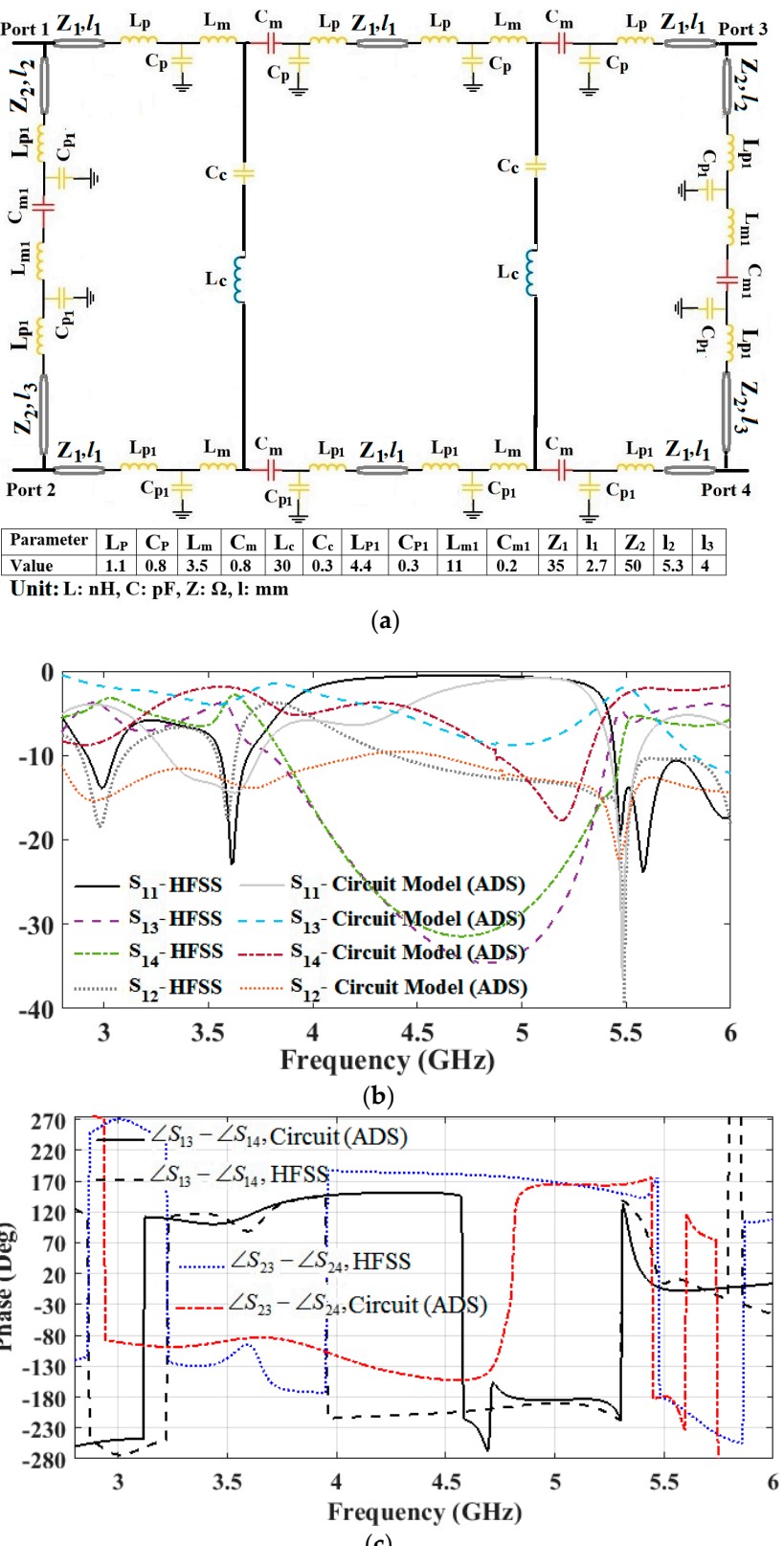

| Parameter | $L_P$ | $C_P$ | $L_m$ | $C_m$ | $L_c$ | $C_c$ | $L_{P1}$ | $C_{P1}$ | $L_{m1}$ | $C_{m1}$ | $Z_1$ | $l_1$ | $Z_2$ | $l_2$ | $l_3$ |
|---|---|---|---|---|---|---|---|---|---|---|---|---|---|---|---|
| Value | 1.1 | 0.8 | 3.5 | 0.8 | 30 | 0.3 | 4.4 | 0.3 | 11 | 0.2 | 35 | 2.7 | 50 | 5.3 | 4 |

**Unit: L: nH, C: pF, Z: Ω, l: mm**

(**a**)

(**b**)

(**c**)

**Figure 5.** (**a**) Accurate circuit model of the proposed coupler with artificial TLs based on OSRR with the obtained elements values, (**b**) amplitude response of the S-parameters of the coupler by using the circuit and EM simulations: (**c**) phase difference of the outputs of the proposed coupler.

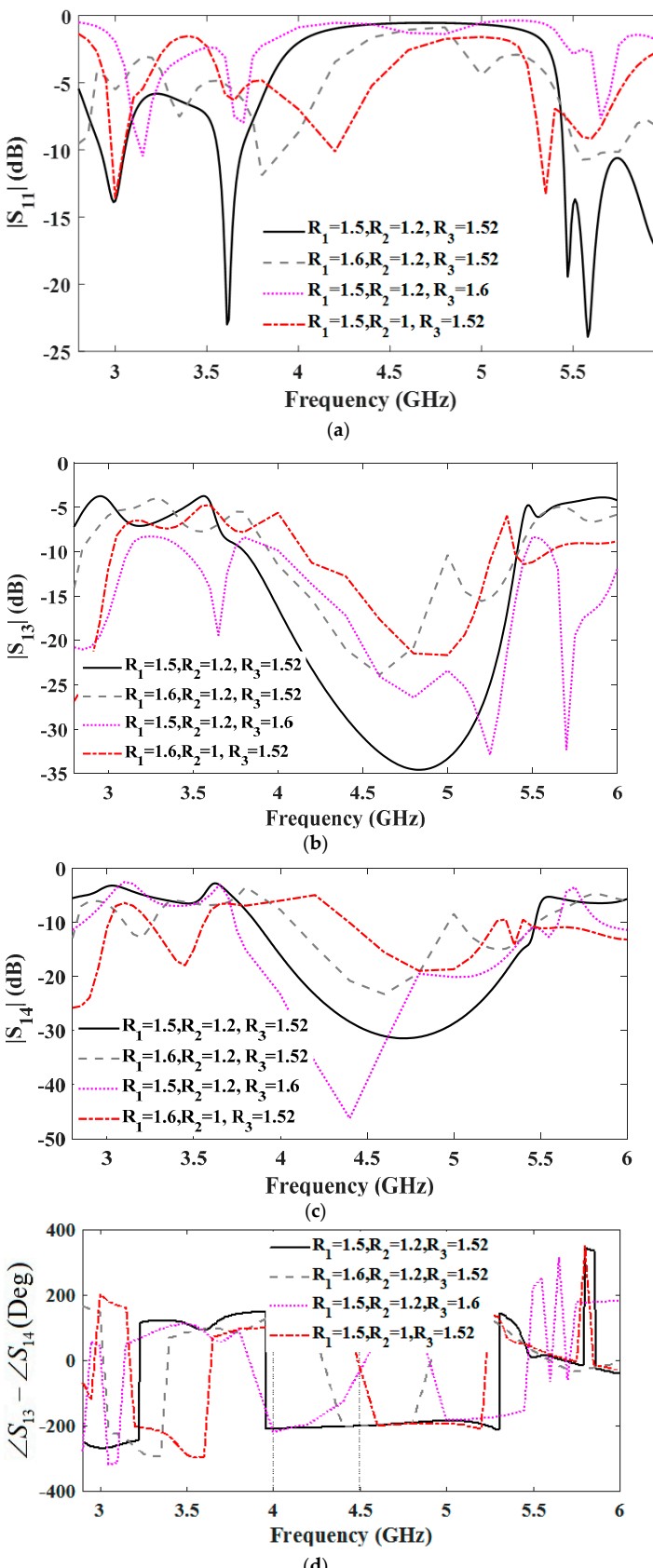

**Figure 6.** The effects of OSRR circular ring radii on the performance of the proposed coupler: (**a**) reflection coefficient, (**b**) transmission coefficient ($S_{13}$), (**c**) transmission coefficient ($S_{14}$) and (**d**) the phase shift difference (the other parameter values are the same as Figure 1).

### 3. Dual-Mode/Dual-band Coupler Simulated and Measured Results

The designed artificial transmission lines were integrated as dual-mode/dual-band couplers. A photograph of the fabricated coupler using the mask etching method is shown in Figure 7. The designed coupler was implemented on a RO4003C substrate with the thickness of 0.8 mm and the dielectric constant of $\varepsilon_r = 3.55$. The overall size of the designed coupler was 13.8 mm × 11.3 mm, corresponding to 0.26 $\lambda_g$ × 0.2 $\lambda_g$ at the first frequency band of operation (where $\lambda_g$ is guided wavelength). The small size of the designed coupler was another interesting aspect.

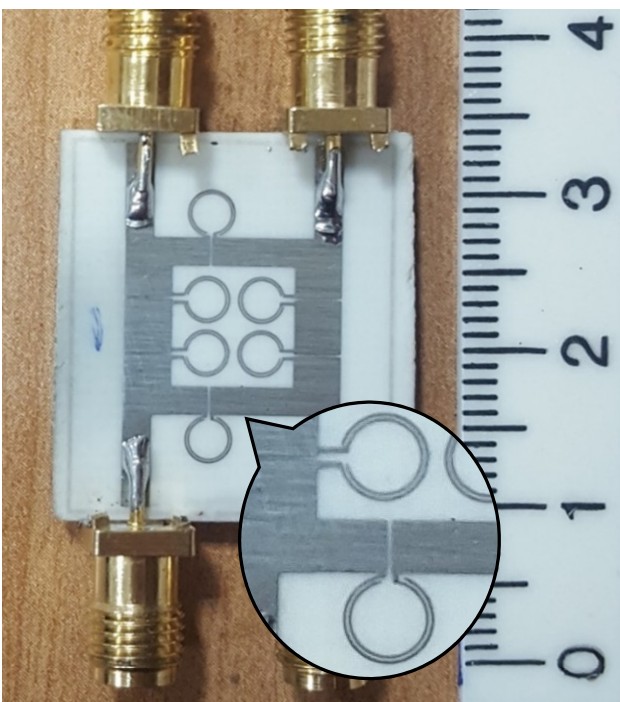

**Figure 7.** Photograph of the fabricated dual-band/mode coupler.

The measured and simulated amplitude and phase responses of the S-parameters are shown in Figure 8. The measurement results were done using the HP8510B network analyzer, and the simulation results were achieved by HFSS software. The measurement results in the branch-line mode were obtained by excitation of port 1. The input impedance matchings of all frequency bands were good and the power splitting ratio (rat-race mod) was acceptable. In the low band, $|S_{31}|$ and $|S_{41}|$ were −4.8/−3 dB and −5/−3.6 dB, respectively, in the dual frequency band of operation. The isolation magnitude responses of the ports were 18/24 dB at the dual frequency of the proposed coupler in the branch line mode. The measured reflection coefficients were −10.5 dB and −21.5 dB for the first and second bands, respectively.

There was approximately a 7% and 5% frequency shift between the simulation and measurement results in the low and high bands. As shown in Figure 6a, by changing the dimensions of the OSRRs, the resonance frequency could be altered. Our fabrication process was reasonably accurate, but it seemed that the variations of the actual dimensions (because of fabrication tolerances) caused a shift in frequency bands of operation. However, as can be seen from the HFSS simulation shown in Figure 8a, the coupler insertion loss increasing in the first band (3.3 GHz) was not due to an improper design. Indeed, losses could be mainly attributed to the ohmic and dielectric losses and possible tolerances in the dimensions of the OSRRs. It should be noted that the structures based on the ring resonators inherently had a narrow bandwidth, but using more unit cells in the proposed configuration could increase the frequency bandwidth at the cost of a larger size.

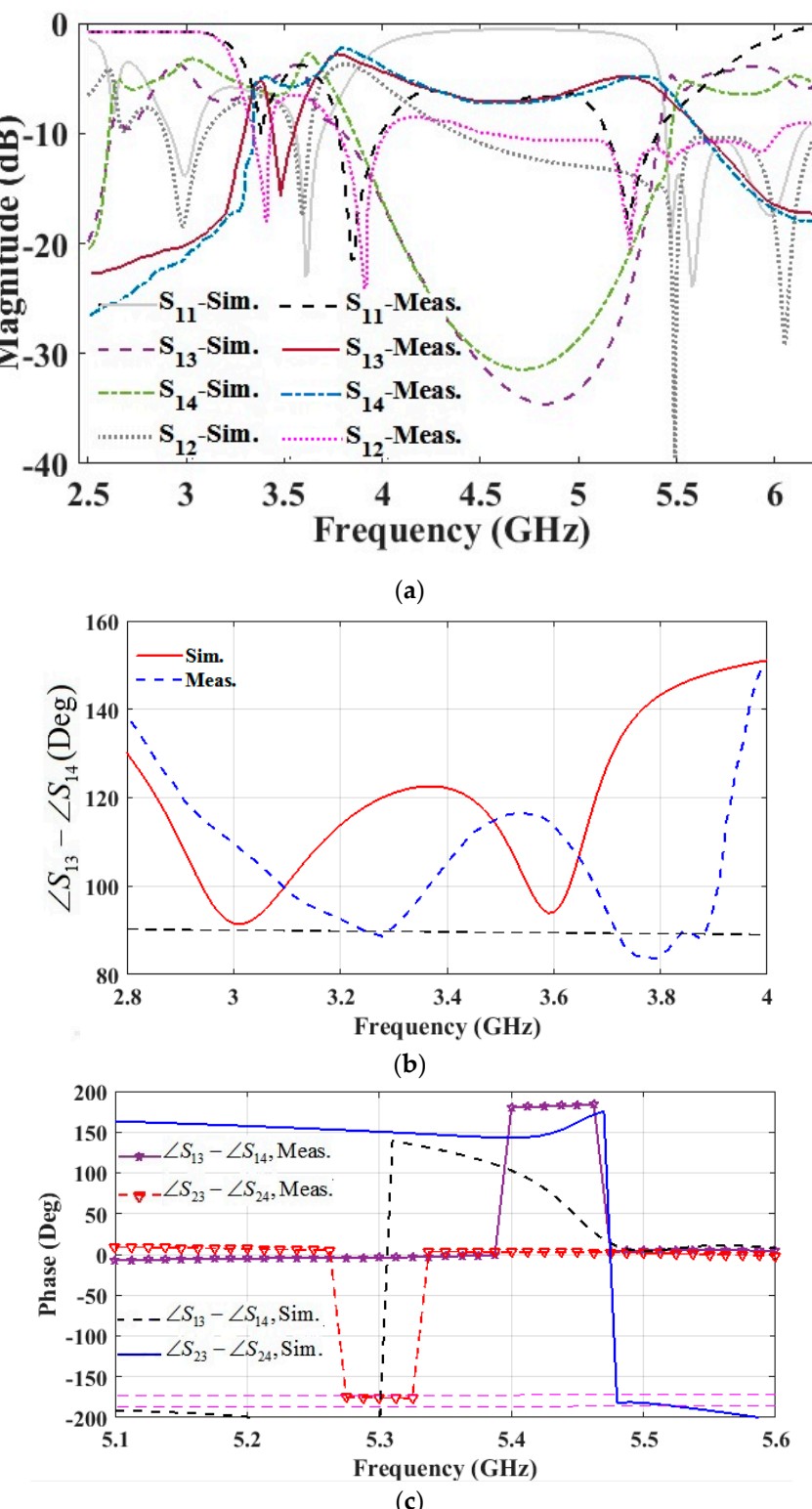

**Figure 8.** Measured and simulated results of (**a**) the reflection and transmission coefficients amplitude response of the proposed coupler, (**b**) phase difference of dual-mode/dual-band coupler in branch line and (**c**) phase difference of the designed coupler in the rat-race.

As we mentioned in the previous section, by considering some circumstances, the rat-race coupler with the outputs on the same side could be realizable. The measured and simulated characteristics of the proposed coupler are demonstrated in Figure 8. The measured return loss was 18.5 dB, the transmission coefficients magnitude ($|S_{13}|$ and $|S_{14}|$)

were -5 dB, and the isolation ($|S_{12}|$) was 20 dB. The simulated and measured power splitting ratios of the proposed coupler in the rat-race mode were between 4 to 5.5 dB. As shown in Figure 2b, the equality between the outputs and power division ratio could be higher, but in the proposed configuration with introduced mutual coupling between the OSRRs, 5 dB power division was achieved.

Figure 8b,c further illustrate the measured and simulated phase differences of the proposed coupler in both modes. As a branch line coupler in the dual frequency band of operation, the measured output differential phases ($\angle S_{13} - \angle S_{14}$) were 88 and 89° at 3.3 and 3.85 GHz center frequencies. In the rat-race coupler, the measured phase differences ($\angle S_{13} - \angle S_{14}$) and ($\angle S_{23} - \angle S_{24}$) were −3.7 and −176°, respectively, at 5.27 GHz, when sum and delta ports were excited. The summary of the performances in different states and frequency bands is illustrated in Table 1.

**Table 1.** Final simulated and measured characteristics of the proposed coupler.

| Prop. Coupler (Mode) | Branch Line (Sim.) | Branch Line (Meas.) | Rat-Race (Sim.) | Rat-Race (Meas.) |
|---|---|---|---|---|
| Center freq. (GHz) | 3.05/3.65 | 3.3/3.85 | 5.45 | 5.27 |
| BW (%) ($|S_{11}| < -10$ dB) | 3.5/4 | 0.5/6 | 3.5 | 4 |
| BW (%) ($|S_{12}| < -15$ dB) | 6.6/2.5 | 2.5/6 | 2 | 4 |
| $|S_{13}|$ (dB) | −3.3/−3 | −4.8/−3 | −5.3 | −5 |
| $|S_{14}|$ (dB) | −4/−3.9 | −5/−3.6 | −5.5 | −5 |
| BW (%) ($|S_{13}| - |S_{14}| < 1.5$ dB) | 8/2 | 2.5/5.5 | 4 | 6 |
| Phase imbalance | 1.5°/2° | 1.5°/1° | 2° | 2.5° |
| BW (%) ($\angle S_{13} - \angle S_{14} < 5°$) | 7/3 | 3/2.5 | 7 | 3 |

A comparison between the proposed circuit performance and other structures in the references is tabulated in Table 2. To our knowledge, there is one structure that could act in dual-mode operation (Reference [3]). The other structures are single dual-band branch line or rat-race coupler while the proposed coupler could be dual-band branch line coupler at one mode and a rat-race coupler at another mode. The dual-band/dual-mode coupler's promising performances in different modes make this coupler a good candidate to be used as the basic part of the feeding structures in a phased array or an arithmetic network for beam switching (Butler matrix or Nolan matrix) or a retrodirective array to reflect the incident signal. Moreover, the proposed configuration could be used in a monopulse circuit as a core building block to miniaturize the entire structure.

**Table 2.** A comparison between the proposed coupler and other references.

| Ref | Size ($\lambda_g^2$) | Frequency (GHz) | Dual Band | Number of Modes (BL and RC) | Coupling (dB) | Phase Imbalance (Deg) |
|---|---|---|---|---|---|---|
| This work | 0.052 | 3.3/3.85/5.27 | YES | 2 | <5 | <2.5 |
| [3] | 0.054 | 1.5/2.5 | YES | 2 | <5 | <5.5 |
| [5] | 0.04 | 0.9/1.8 | YES | 1 | <5 | <2.5 |
| [6] | 0.091 | 3.1/5 | YES | 1 | <4.3 | <4 |
| [8] | - | 0.9/2 | YES | 1 | <3.5 | <5 |
| [10] | 0.162 | 0.9/1.8 | YES | 1 | <4.1 | <1 |
| [12] | - | 1/2 | YES | 1 | <3.7 | <1 |
| [13] | - | 4.3/7.6 | YES | 1 | <3.9 | - |
| [14] | 0.017 | 0.87/1.79 | YES | 1 | <3.9 | <2 |
| [17] | - | 2.45/6.1 | YES | 1 | <4.8 | <1 |
| [19] | 0.072 | 1/5.2 | YES | 1 | <4.3 | <1.5 |
| [20] | 0.042 | 0.7/2.55 | YES | 1 | <3.5 | <2 |

$\lambda_g$: is the guided wavelength at the lowest operating frequency. BL: branch line, RC: rat race.

## 4. Conclusions

A novel compact dual-band/dual-mode coupler was proposed to operate as a dual-band branch line coupler at the low frequency bands (3.3 GHz and 3.85 GHz) and a 180° rat-race coupler in the high frequency band (5.3 GHz). We designed a rat-race coupler with the outputs at the same side and created a multi-band operation with different characteristics. The coupler was implemented using artificial transmission lines loaded with different OSRRs. The characteristic impedance and insert phase of the artificial TLs at three different frequencies were designed and modelled. The theoretical analysis and synthesis of OSRR TL were performed to provide a design guideline. The obtained phase imbalance was less than 2.5°, as well as the insertion loss which was less than 2 dB across the whole frequency band. The desirable performance of the proposed coupler in branch-line and rat-race modes make it an excellent choice to be used as a core building block in feeding or arithmetic networks of antenna arrays for beam steering or direction-finding applications.

**Author Contributions:** M.S.G. designed and fabricated the proposed structure. M.S.G. and S.K. measured the couplers and wrote the manuscript. M.M.H. and R.M. contributed to the discussion and reviewed the manuscript. G.M. supervised the project and contributed to discussion, and reviewed the manuscript. All authors have read and agreed to the published version of the manuscript.

**Funding:** This research received no external funding.

**Conflicts of Interest:** The authors declare no conflict of interest.

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
