# Peer review of "Dual-Band/Dual-Mode Rat-Race/Branch-Line Coupler Using Split Ring Resonators"

_electronics, doi:10.3390/electronics10151812_

Round 1
Reviewer 1 Report
The topic of this paper is interesting to the readers, within the scope of the journal, however prior its publication changes must be done.
The use of English must be improved. The paper includes several grammatical and syntax errors.
The majority of the figures are not clear enough. Increase the size and improve the quality of them.
The Introduction must be revised. The authors must present the general research area to unfamiliar readers and at most to present the current state-of-the-art in order to show the contribution/novelty of their work. Authors must describe/analyse more the current mentioned references and must include many more related references, such as the following:
Vitas A., Vita V., Chatzarakis G.E., Ekonomou L., Review of different ring resonator coupling methods, Proceedings of the 9th WSEAS Int. Conf. on Telecommunications and Informatics (TELE-INFO ’10), Catania, Sicily, Italy, May 29-31, pp. 227-231, 2010.
Ekonomou L., Vita V., Chatzarakis G.E., Testing of a microwave transmission link system at 2.45GHz, Proceedings of the 9th WSEAS Int. Conf. on Data Networks, Communications, Computers (DNCOCO '10), University of Algarve, Faro, Portugal, pp. 21-25, 2010.
A separate discussion section that will comment on the produced results must be included.
Conclusions must summarize the work presented within the paper.
Author Response
--Thanks for your valuable comments. The written paper is carefully revised to fix grammatical errors. The quality of entire figures is improved in the whole paper.
--The introduction section is amended and some of the new papers is added to the manuscript. Thank you for your recommended paper.
--A performance comparison table is added to the manuscript to compare the obtained results with other references.
--The conclusion section is amended to demonstrate the whole paper summary.

Reviewer 2 Report
In this paper, authors present a compact hybrid coupler using artificial transmission lines for dual-band operation. The proposed structure is analyzed theoretically using microwave theory. Its performance is verified using by simulation and measurements. The paper is well organized. The reviewer suggests the comments as follows.
- It would be better if the advantage and limitation of the proposed structure were compared with previous works. Please review the previous works and add the table of performance comparison in the revised manuscript.
Author Response
Thanks for your comments. A comparison table is added to the paper to demonstrate the proposed coupler performance in comparison with other structures in the literature.
Reviewer 3 Report
This manuscript has a very high proportion of similarities (about 70-80%) with the authors' earlier published article, Dual band/ Dual Mode Branch-Line/Rat-Race Coupler Using Artificial Transmission Line, 2019 27th Iranian Conference on Electrical Engineering (ICEE), 2019, pp. 1622-1626, as cited in reference [4] in the manuscript.
Please highlight the additional and significant contribution of this manuscript as compared with the earlier published work.
Without adding more work and highlight the significance, the manuscript will be rejected in its current form.
Author Response
Thanks for reviewer comments and consideration. The proposed paper is a complete extension of the author’s short paper in the ICEE conference. We just presented the full paper in this submission. The proposed paper contains a huge analytical and circuit model simulation to show how the proposed coupler works at the different modes as well as parametric study to investigate the proposed coupler parameters. The circuit model (Section 2-A, 2-B, 2-D and 2-E and parametric analysis (Section 3) are new materials in this submission. We also added a comparison table to compare the proposed coupler characteristics with other circuits in the literature.
It should be noted that the conference paper consists of about 3000 words and 6 plots while the current paper includes approximately 5500 words and 18 plots and 2 tables. So, there is a significant difference (more than 60%) between these papers.
Round 2
Reviewer 1 Report
The authors did not fully complied with reviewers' comments.
E.g., The introduction must be enriched with the inclusion of more references in order to be clear the current state-of-the-art and the contribution/novelty of the current work.
A separate discussion section that will comment on the produced results must be included.
Conclusions must summarize the work presented within the paper.
Author Response
Thanks for the reviewer comments. The manuscript revised again to full fit reviewer comments. The revised sections are highlighted through the paper.
1- The introduction section is amended with more references. As mentioned in the paper and can be seen in the comparison table, there are a few couplers in the articles which could operate as a dual mode branch line and rat-race couplers, the other structures are usually dual band or multi band not dual mode. So, the novelty of the proposed design is based on dual-mode and dual-band operation in one mode which is presented in this article for the first time. We tried to demonstrate the novelty of current work as compared to other structures in the introduction section and comparison table ( Table. 2).
2- The measurement and simulation results are specifically discussed and investigated in section 3. By considering the achieved results, the proposed design characteristics are tabulated with details in Table. 1. Also, the simulation and measurement results are compared together and the reason for differences between them is explained in this section. So, the results and discussion on them are presented clearly in a separate section of the paper (Section. 3).
3- All work presented in the paper such as results, analysis, circuit models and simulation methods are included in the conclusion now. So, the authors think that the conclusion summarizes the article results.

Reviewer 3 Report
The paper is an extension of a conference paper. The authors have increased the content for completeness. No further comments from me.
Author Response
Thanks for your comments.